# Interaction Force Experiments for Force-Guided Assembly.

*Abstract*— Insertion tasks, such as cable and electrical connector insertion, are performed primarily by humans on assembly lines. The flexibility of the cables and wires, and the tight-tolerance insertions, with likely visual occlusions, make the automation of these tasks difficult. Force-guided insertion is a robust alternative for performing insertion tasks under occlusion conditions. However, the capabilities of force-guided approaches are compromised when there is no proper alignment of the object with the hole/connector. In this note, we aim to infer orientation and/or position errors in the object alignment using only force and moment measurements, which can be useful for corrections during force-guided insertion. We analyze the interaction forces and moments acting during the insertion of a peg/cable into a tight hole with a diameter approximately equal to the peg. The forces are recorded using a UR5e robot with a Robotiq Hand-E parallel gripper (with a force/torque sensor) grasping a cable/peg rigidly, and a hermetic pass-through with a hole of the same diameter as the cable/peg. Several insertion tests were performed considering rotation and position alignment errors. The experimental results present analogous behaviour of the force in the direction of the insertion, and the other two orthogonal forces, no matter if there is orientation or position error. Then, the moments were considered, and a hint about the rotation error was found. The torque measurements and their principal component analysis (PCA) show a completely different direction when positive and negative rotation errors occur.

## I. INTRODUCTION

Object insertion is a central task in assembly operations, executed mainly by hand. The automation of insertion is challenging since the task has to be executed in unstructured environments, where the object position and orientation are difficult to estimate [1].

Methods based on force and visual sensors, and their fusion, have been presented to address object insertion. A wire is inserted in a hole in [2] using a vision-based approach. In [3], a multimodal model based on vision and force perception is used to design a compliant control that accomplishes the insertion of a cable of into a square gap. The insertion of a male type-C connector into a female type, grasping the cable (not the connector), is achieved in [4] using visual sensing. A simulation of the insertion of a deformable peg with a rigid hole is reported in [5]. However, most of the techniques are based on visual feedback, and occlusion is likely to occur in real-life insertion scenarios.

Force-guided insertion based only on force and torque measurements is a robust approach to accomplish insertion tasks in the presence of occlusion [6]. However, the potential of force-guided insertion depends on a correct alignment between the object and the hole.

This note presents laboratory tests of the interaction forces and moments acting during the insertion of a peg/cable into a tight hole whose diameter is approximately equal to the peg/cable diameter. Our target is to find useful information for force-guided assembly in the force and moment measurements, such as error alignments (orientation and position). The laboratory test showed that the measurements corresponding to the force in the direction of the insertion do not show much information about position or orientation errors, since no matter what error is tested, these signals present analogous behaviour. Therefore, the moments were used, and their data gave a hint about orientation errors.

## II. TEST DESCRIPTION

The tests consist of passing a wooden stick and cable through a tight hole, see Figure 1. We consider the stick/cable is rigidly grasped without slip. The tests start with the stick/cable aligned with the hole, i.e., reference frame $O_{x_g,y_g,z_g}$ is aligned with the reference frame $O_{x_c,y_c,z_c}$. Then a linear motion along $y_g$ is executed to pass the stick through the hole.

Once the first insertion is done, rotation errors are added with increments of five degrees around $x$ and $z$. The position errors are added as increments of one millimeter along $x$ and $z$. After each error increment, the same linear movement along $y_g$ is executed to achieve the insertion. For the case of the cable, a bend upwards and downwards at the tip was also considered. During the test, the forces and moments are measured with respect to the gripper reference frame $O_{x_g,y_g,z_g}$, see Figure 1.

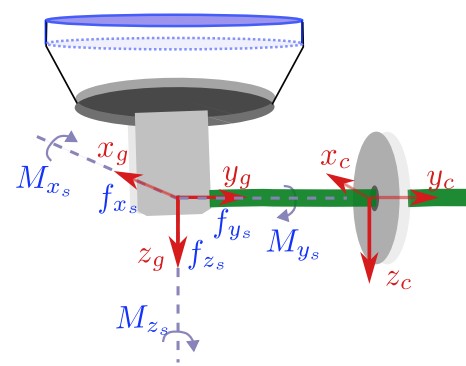

Fig. 1. Reference frame at the gripper $O_{x_g,y_g,z_g}$, at the hole $O_{x_c,y_c,z_c}$. Forces and torques are measured with respect to the gripper reference frame $O_{x_g,y_g,z_g}$

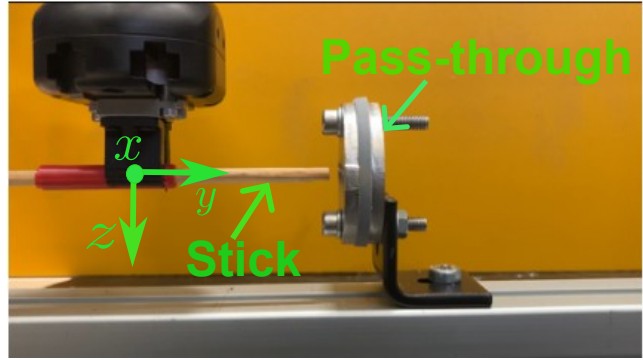

Fig. 2. Experimental setup: gripper, wooden stick, and pass-through.

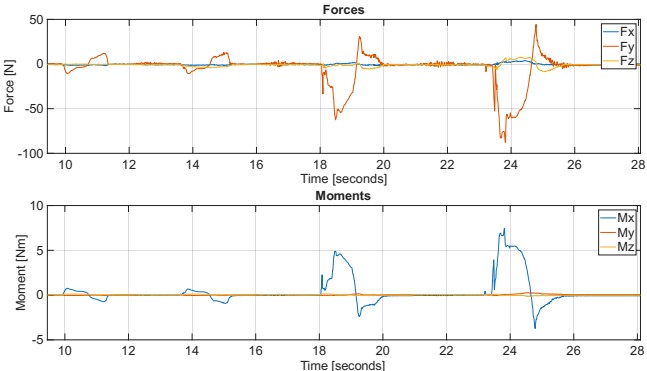

Fig. 3. Forces and moments during insertion: rotation errors around $x$. The four observed transients correspond with the cases of no error, 5, 10, and 15 degrees of error.

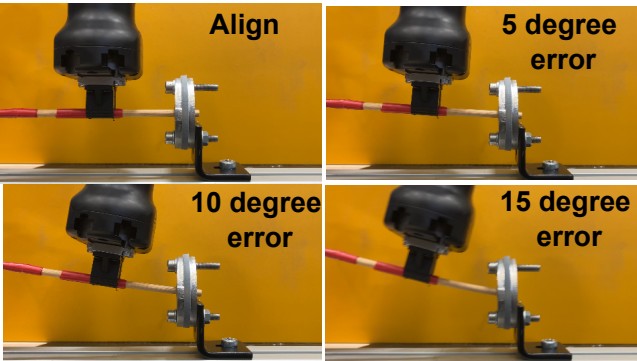

Fig. 4. Video frames of insertion test with rotation error around $x$.

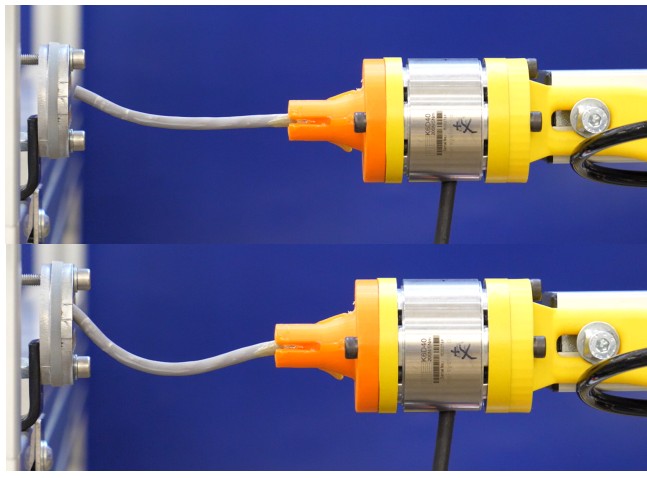

Fig. 5. Bending of the cable around $x$. Top: Before insertion trial. Bottom: after insertion trial

## III. LABORATORY TESTS

The experimental setup consists of a UR5e robot with a Robotiq Hand-E parallel gripper grasping a 6 mm diameter stick/cable, and a hermetic pass-through with a 6 mm diameter hole, see Figure 2.

Considering position and orientation errors in $x$ and $z$, and around $x$ and $z$, respectively, and bending for the cable. The interaction forces measured at the grip's reference frame are recorded.

All the tests have the following sequence of linear movements. The stick/cable is aligned with the hole. The cable is inserted in the hole. And, the cable is returned to the aligned position. Then, a position error (rotation error) of one millimeter (five degrees) in (around) $x$ or $z$ is added, and the sequence is repeated.

Figure 3 shows the interaction forces corresponding to the tests with rotation error around the $x$ axis, see the video frames of this test in Figure 4. One can see that the biggest force and moment magnitudes are along the $y$ direction and around the $x$ axis, respectively, since the insertion direction is along the $y$ axis.

Note that analogous behavior (biggest moment around $x$ and biggest force along $y$) is observed when errors around $z$, and along $x$ and $z$ were tested. Similar tests were done using a flexible cable (see video-link), where you can observe similar behavior that is presented in Figure 3.

Considering the cable is bent before insertion, see Figure 5, a second set of tests was performed. The bending of the cable can be considered as a rotational error, for example, a rotation error around the $x$ axis is seen in Figure 5.

The plot of torques measured during a set of tests with positive and negative rotational error (bending) around $x$ is presented in Figure 6, where G1 and G2 labels refer to negative and positive rotation error, respectively. The forces are not presented since they shown analogues behavior than Figure 3.

From Figure 6, one can observe that the torques around $x$ and $y$ show positive and negative values when the rotation error is negative and positive, respectively. A principal component analysis (PCA) of the torque data with the first eigenvectors plotted for both errors around $x$, with the mean of the vectors plotted in bold, is presented in Figure 7. One can observe that the bold vectors' direction shows a different direction for the negative and positive error cases, which concur with the information provided by the measured torques. Then, a clue about the rotation error can be obtained from the torque measurements.

## IV. COMMENTS AND FUTURE WORK

The tests presented in this note are the first trials towards an implementation of force-guided insertion in the assembly

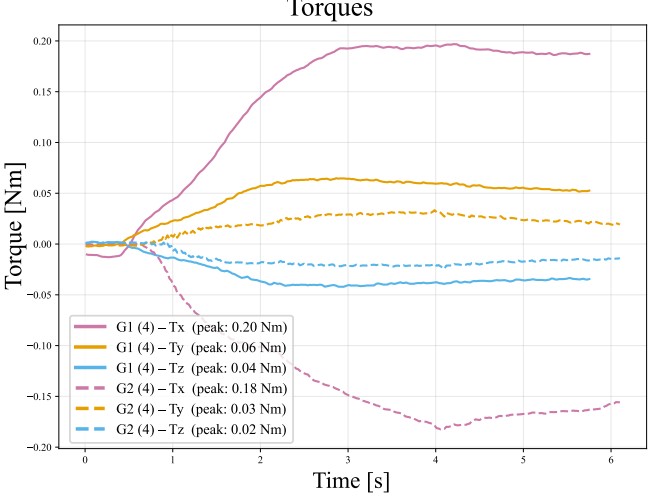

Fig. 6. Bending of the cable around $x$. Top: Before insertion trial. Bottom: after insertion trial

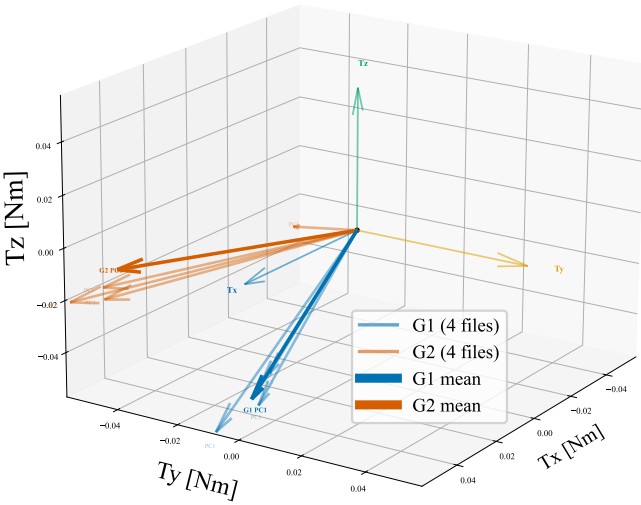

Fig. 7. Principal component analysis of torque measurements. G1 and G2 labels refer to negative and positive rotation error, respectively. The mean of the vectors is plotted in bold.

automation of devices that require a gas(liquid)-tight seal, such as electrolyzers, vacuum furnaces, gloveboxes, etc. That's why the insertion is done using a hermetic feed trough in the insertion test.

The next stage is a detailed analysis of the recorded forces and torques using exploratory data tools, such as Principal Component Analysis (PCA).

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
