# OpenReview forum: "Interaction Force Experiments for Force-Guided Assembly"
_IEEE.org/ICRA/2026/Workshop/Manipulation_Robustness — ICRA 2026_

### Official Review · Reviewer_KySk · 2026-04-30
**Clear strengths in problem relevance, but weaknesses in experimental transparency**

**Rating:** 6
**Confidence:** 4

**Review:**

**Overall comments：**

This paper presents a set of preliminary experimental observations on force/torque signals during tight-tolerance insertion tasks. The empirical finding that torque directions differ between positive and negative rotational errors, supported by PCA visualization, is an interesting observation. The use of a real robot and both rigid and flexible objects is a strength.

**Strengths ：**

The paper addresses a real industrial challenge: tight-tolerance insertion under visual occlusion using only force/torque sensing. This is highly relevant to force-guided assembly in applications such as hermetic feed-throughs for electrolyzers or vacuum furnaces.

 **Weaknesses ：**

- The number of repetitions per error condition is not specified.
- No statistical summaries (e.g., mean ± std, box plots) are provided.
- The video link is mentioned but not included in the manuscript.

---

### Decision · Program_Chairs · 2026-05-21

Accept